# Square Root Principal Component Pursuit: Tuning-Free Noisy Robust Matrix Recovery

**Junhui Zhang**
Department of Applied Physics and Applied Math
Columbia University
New York, NY 10027
jz2903@columbia.edu

**Jingkai Yan**
Department of Electrical Engineering
Columbia University
New York, NY 10027
jy2927@columbia.edu

**John Wright**
Department of Electrical Engineering
Columbia University
New York, NY 10027
jw2966@columbia.edu

## Abstract

We propose a new framework – Square Root Principal Component Pursuit – for low-rank matrix recovery from observations corrupted with noise and outliers. Inspired by the square root Lasso, this new formulation does not require prior knowledge of the noise level. We show that a single, universal choice of the regularization parameter suffices to achieve reconstruction error proportional to the (a priori unknown) noise level. In comparison, previous formulations such as stable PCP rely on noise-dependent parameters to achieve similar performance, and are therefore challenging to deploy in applications where the noise level is unknown. We validate the effectiveness of our new method through experiments on simulated and real datasets. Our simulations corroborate the claim that a universal choice of the regularization parameter yields near optimal performance across a range of noise levels, indicating that the proposed method outperforms the (somewhat loose) bound proved here.

## 1 Introduction

The problem of recovering a low-rank matrix from unreliable observations arises in a wide range of engineering applications, including collaborative filtering [1], latent semantic indexing [2], image and video analysis [3, 4, 5] and so on. This problem can be formalized in terms of the following observation model: given an observation $D$ which is a superposition

$$D = \underset{\text{low-rank}}{L_0} + \underset{\text{sparse}}{S_0} + \underset{\text{noise}}{Z_0}. \tag{1.1}$$

35th Conference on Neural Information Processing Systems (NeurIPS 2021).

of an unknown low-rank matrix $L_0$, sparse corruptions $S_0$ and dense noise $Z_0$, our goal is to accurately estimate both $L_0$ and $S_0$.

This model has been intensely studied, leading to algorithmic theory for methods based on both convex and nonconvex optimization [6, 7, 8]. One virtue of the convex approach is that in the noise-free setting ($Z_0 = 0$), it is possible to exactly recover a broad range of low-rank and sparse pairs ($L_0, S_0$), with a universal choice of regularization parameters, which does not depend on either the rank or sparsity. This makes it possible to deploy this method in a "hands-free" manner, provided the dataset of interest indeed has low-rank and sparse structure.

In the presence of noise, however, the situation becomes more complicated: all efficient, guaranteed estimators require knowledge of the noise level (or the rank and sparsity) [8, 9, 10]. This is problematic, since in most applications the noise level is not known ahead of time. In standard convex formulations, the appropriate regularization parameter depends on the noise standard deviation, leaving the user with a painful and time-consuming task of tuning these parameters on a per-dataset basis.

Motivated by this issue, we revisit this classical matrix recovery problem. The main contribution of this paper is the proposal and analysis of a new formulation for robust matrix recovery, which stably recovers $L_0$ and $S_0$ without requiring prior knowledge of the rank, sparsity, or noise level. In particular, our approach admits a single, universal choice of regularization parameters, which under standard hypotheses on $L_0$ and $S_0$, yields an estimation error proportional to the noise standard deviation $\sigma$. To our knowledge, our method and analysis are the first to achieve this.

Our approach is based on a combination of two natural ideas. For matrix recovery, we draw on the *stable principal component pursuit* [9], a natural convex relaxation, which minimizes a combination of the nuclear norm of $L$, the $\ell_1$ norm of $S$ and the squared Frobenius norm $\|Z\|_F^2$ of the noise. This is a principled approach to handling both the structured components $L_0, S_0$ and the noise: $\|Z\|_F^2$ can be motivated naturally from the negative log-likelihood of the gaussian distribution. Moreover, under mild assumptions on the rank and singular vectors of $L_0$ and the sparsity pattern of $S_0$, the reconstruction error of stable PCP is $O(\|Z_0\|_F)$ [9].

On the other hand, optimally balancing these terms requires knowledge of the standard deviation $\sigma$ of the true noise distribution. To address this issue, we draw inspiration from the square root Lasso [11]. The square root Lasso is a sparse estimator which achieves minimax optimal estimation with a universal choice of parameters, which does not depend on the noise level. The core idea is very simple: instead of penalizing the squared Frobenius norm $\|Z\|_F^2$ of the noise, one penalizes its square root, $\|Z\|_F$. We call the resulting formulation *square root principal component pursuit* ($\sqrt{\text{PCP}}$). Our new formulation has the benefit that with a *noise-independent* universal choice of regularization parameters, essentially the same level of reconstruction error can be achieved. This makes $\sqrt{\text{PCP}}$ a more practical approach to low-rank recovery in unknown noise.

Due to the square root term, the objective function is no longer smooth or differentiable, and so we cannot apply algorithms such as the proximal gradient method[1]. Nevertheless, our new formulation remains convex and separable, i.e. the objective is the sum of functions of different variables, making Alternating Direction Method of Multipliers (ADMM) a suitable solver [12]. We test our new formulation with ADMM on both simulated data and real data in image processing. The experimental results show the effectiveness of our proposed new formulation in recovering the low-rank and the sparse matrix, and suggest that $\sqrt{\text{PCP}}$ has better performance than anticipated by our loose upper bound of the reconstruction error.

## 1.1 Notations and Assumptions

We use $\|A\|$, $\|A\|_F$, and $\|A\|_*$ to denote the spectral, Frobenius, and nuclear norm of the matrix $A$, and $A^*$ its (conjugate) transpose. For convenience, we let $X_0 = (L_0, S_0)$ be the concatenation of $L_0$ and $S_0$. We assume that $L_0$ is a low-rank matrix of rank $r$ whose compact

---

[1] which requires the objective to be the sum of a smooth and a non-smooth function

SVD is $L_0 = U\Sigma V^*$, $U \in \mathbb{R}^{n_1 \times r}$, $V \in \mathbb{R}^{n_2 \times r}$, and without loss of generality, $n_1 \geq n_2$. Let $T = \{UQ^* + RV^* | Q \in \mathbb{R}^{n_2 \times r}, R \in \mathbb{R}^{n_1 \times r}\}$ denote the tangent space of rank $r$ matrices at $L_0$. In addition, we assume that $S_0$ is sparse with support in $\Omega$.

Since it is impossible to disentangle $L_0$ and $S_0$ if the low-rank matrix $L_0$ is sparse, or if the sparse matrix $S_0$ is low-rank, we make the following two assumptions:

**Assumption 1.1** *The low-rank matrix $L_0$ satisfies the incoherence property with parameter $\nu^2$, i.e.*

$$\max_i \|U^* e_i\|^2 \leq \frac{\nu r}{n_1}, \quad \max_i \|V^* e_i\|^2 \leq \frac{\nu r}{n_2}, \quad \|UV^*\|_\infty \leq \sqrt{\frac{\nu r}{n_1 n_2}}.$$

**Assumption 1.2** *The support $\Omega$ is chosen uniformly among all sets of cardinality $m$, and the signs of supports are random, i.e. $P[(S_0)_{i,j} > 0 | (i,j) \in \Omega] = P[(S_0)_{i,j} < 0 | (i,j) \in \Omega] = 0.5$.*

These assumptions follow [6]; indeed, our proof makes use of a dual certificate constructed for noiseless low-rank and sparse recovery in that paper.

## 1.2 Problem Formulation and Main Results

Inspired by square root Lasso, we propose to solve the robust noisy matrix recovery problem through the following optimization problem:

$$\sqrt{\text{PCP}} : \min_{L,S} \|L\|_* + \lambda \|S\|_1 + \mu \|L + S - D\|_F. \tag{1.2}$$

The parameter $\lambda$ that balances the low-rank and the sparse regularizers is studied in [6], where it is shown that $\lambda = 1/\sqrt{n_1}$ gives exact recovery when $Z_0 = 0$ and the $\mu \|L + S - D\|_F$ penalty term in (1.2) is replaced with the constraint $L + S = D$. In this work, we build on this result and focus on the parameter $\mu$. Our main result is that under the aforementioned (standard) hypotheses on $L_0$ and $S_0$, using a single, universal choice $\mu = \sqrt{n_2/2}$, $\sqrt{\text{PCP}}$ recovers $L_0$ and $S_0$, with an estimation error that is proportional to the norm of the noise:

**Theorem 1.1** *Under Assumptions 1.1 and 1.2, provided that $r, m$ satisfies*

$$r \leq \rho_r n_2 \nu^{-1} (\log n_1)^{-2}, \quad m \leq \rho_s n_1 n_2, \tag{1.3}$$

*where $\rho_r \leq 1/10$, $\rho_s$ are some positive constants. Then there is a numerical constant $c$ such that with probability at least $1 - cn_1^{-10}$, the $\sqrt{\text{PCP}}$ problem (1.2) with $\lambda = 1/\sqrt{n_1}$ and $\mu = \sqrt{n_2/2}$ produces a solution $\widehat{X} = (\widehat{L}, \widehat{S})$ such that*

$$\|\widehat{X} - X_0\|_F \leq 560 \sqrt{n_1 n_2} \|Z_0\|_F. \tag{1.4}$$

Why is it possible to achieve accurate estimation with a single choice of $\mu$? We draw intuition from a connection to the *stable principal component pursuit* formulations studied in [9]. This work studies both constrained and unconstrained formulations:

$$\text{StablePCP}_c : \min_{L,S} \|L\|_* + \lambda \|S\|_1 \; s.t. \; \|L + S - D\|_F \leq \delta. \tag{1.5}$$

$$\text{StablePCP}_u : \min_{L,S} \|L\|_* + \lambda \|S\|_1 + \frac{\bar{\mu}}{2} \|L + S - D\|_F^2. \tag{1.6}$$

These formulations are equivalent, and equivalent to $\sqrt{\text{PCP}}$ in the following sense: for each problem instance, there is a calibration of parameters $\delta \leftrightarrow \bar{\mu} \leftrightarrow \mu$ such that $\sqrt{\text{PCP}}$, $\text{StablePCP}_u$ and $\text{StablePCP}_c$ have exactly the same set of optimal solutions. However, (1.5)-(1.6) require that the parameters $\delta$ and $\bar{\mu}$ be determined on an instance-by-instance basis, based on the noise level. Choosing these parameters correctly is essential: in (1.5), $\delta$ should be chosen to be larger than $\|Z_0\|_F$. For square ($n_1 = n_2 = n$) matrices, in a stochastic setting in which $(Z_0)_{ij}$ are iid $\mathcal{N}(0, \sigma^2)$, following [9], $\bar{\mu}$ can be chosen as $\frac{1}{2\sigma\sqrt{n}}$. This follows from

---

[2] $\nu \geq 1$ since $\|U\|_F^2 = r$, and so $\max_i \|U^* e_i\|^2 \geq \frac{r}{n_1}$.

the fact that in this setting $n^{-1/2}\|\boldsymbol{Z}_0\| \to 2\sigma$ almost surely; setting $\bar{\mu}$ in this fashion ensures that the singular value shrinkage induced by the nuclear norm regularizer $\|\cdot\|_*$ is greater than the largest singular value of $\boldsymbol{Z}_0$.

In contrast to this $\sigma$-dependent penalty parameter, fixing $\boldsymbol{S} = \boldsymbol{S}_0$, $\sqrt{\text{PCP}}$ formulation (1.2) requires that $\boldsymbol{0} \in \partial(\|\widehat{\boldsymbol{L}}\|_* + \mu\|\widehat{\boldsymbol{L}} - \boldsymbol{Z}_0 - \boldsymbol{L}_0\|_F)$, which translates into $-\mu\frac{\widehat{\boldsymbol{L}} - \boldsymbol{Z}_0 - \boldsymbol{L}_0}{\|\widehat{\boldsymbol{L}} - \boldsymbol{Z}_0 - \boldsymbol{L}_0\|_F} \in \partial\|\widehat{\boldsymbol{L}}\|_*$. With the hope that $\widehat{\boldsymbol{L}} \approx \boldsymbol{L}_0$, and by the subdifferential formulation[3], we have $\mu\frac{\|\boldsymbol{Z}_0\|}{\|\boldsymbol{Z}_0\|_F} \approx 1$. The concentration stated above then gives an intuitive choice for $\mu \approx \frac{n\sigma}{2\sigma\sqrt{n}}$, or $\mu = c_0\sqrt{n}$ for some $c_0 > 0$. The magic of $\sqrt{\text{PCP}}$ is that by using the Frobenius norm instead of its square, the objective function becomes homogeneous, i.e. the gradient of the penalty term at the ground truth $\boldsymbol{X}_0$ becomes $\sigma$ independent, making a universal penalty parameter possible.

### 1.3 Relationship to the Literature

The problem of low-rank matrix recovery from gross sparse corruption can be considered a form of robust PCA, [6, 13], and has been studied extensively in the literature. Algorithmic theory has been developed for both convex [6, 13, 14, 15, 16], and nonconvex optimization methods [7, 8, 17, 18, 19]. While many of the aforementioned works pertain to noiseless data, a line of work has studied extensions to noisy data. [9] studied the problem of robust matrix recovery with bounded noise under the incoherence assumption, and proved a bound on the recovery error, with linear dependence on the noise level but suboptimal dependence on the matrix size. [20] studied the problem with a weaker assumption about spikiness using decomposable regularizers and restricted strong convexity (RSC), and obtained essentially optimal bounds on the reconstruction error when the noise level is large. These weaker assumptions are not sufficient to ensure exact recovery, and so when the noise standard deviation $\sigma$ is small, this approach does not yield a reconstruction error proportional to the noise level. [10] formulated robust PCA as a semidefinite programming problem, which requires strong assumptions about square matrices and positive semidefiniteness of the low-rank matrix. Some other works [21, 22] further assumed partial observation of the matrix, and also derived tighter bounds on the recovery error. The recent work of [8] achieves optimal error bounds for both large and small noise, using a novel analysis that leverages an auxiliary nonconvex program. Taken together, these results give efficient and provably effective methods, whose statistical performance is nearly optimal. Compared to e.g., [8, 20], the stability guarantees provided by our theory are worse by a dimension-dependent factor. Nevertheless, all of the above works regarding robust PCA with noise, the optimization involves parameters that must be set based on the noise level distribution, and therefore challenging in actual applications.

On the other hand, there has been existing work in the literature on structured signal recovery without needing to know the noise level. Our proposed square root PCP is directly inspired by the square root Lasso [11], which proposed the idea of replacing the squared loss with its "square root" version. This allows for a choice of the parameter independent of noise level, while maintaining near-oracle performance. Later works have extended this idea to other scenarios, such as group lasso [23], SLOPE variable selection [24], elastic net [25] and matrix completion [26], etc. Notably, the work of [26] studied the matrix completion where one aims to recover a low-rank matrix from noisy linear observations, aka matrix completion. Compared with that paper, this work aims to solve a different problem where the observation also contains a sparse outlier matrix. To the best of our knowledge, this paper is the first to propose a provable algorithm for Robust PCA with noisy observation that does not require knowledge of the noise level beforehand. On the algorithmic side, interior point method and first order method are used to solve the square root Lasso in [11], while later works apply ADMM to the problem [27][28]. In our problem, the objective function can be transformed into a separable form, making ADMM a reasonable choice.

We note that for large $\sigma$ the error bound established in this paper is suboptimal compared with [11] and [26]. The problem of square root lasso enjoys benign properties (lower bounds on restricted eigenvalues) which we do not have in square root PCP. The paper [26] on matrix completion makes a spikiness assumption, and proves that a square root

---

[3]The subdifferential of a norm satisfies $\partial\|x\| = \{z \mid \langle x, z \rangle = \|x\|, \|z\|^* \leq 1\}$.

lasso-inspired formulation achieves essentially optimal estimation when the noise is large. As with robust matrix recovery, the spikiness assumption is not strong enough to imply exact recovery in the noiseless case. Compared to these works, the principal differences in this paper are (i) the problem formulation: we consider robust PCA with sparse errors, (ii) the analysis, which proceeds down different lines, and (iii) that our bounds are linear in the noise level, for both large and small noise. However, in contrast to [26], our analysis does not yield minimax optimal estimation errors; it is worse by a dimension-dependent factor. Improving this dependence is an important direction for future work.

## 2    Analysis

The proof of the main Theorem 1.1 is different from the standard approach in [11] due to a lack of the Restricted Strong Convexity property for the map $(\boldsymbol{L}, \boldsymbol{S}) \to \|\boldsymbol{L} + \boldsymbol{S} - \boldsymbol{D}\|_F$. Instead, our approach has three key ingredients:

- The result from StablePCP$_c$ (Theorem 2.1) shows a recovery error $\|(\widehat{\boldsymbol{L}}, \widehat{\boldsymbol{S}}) - (\boldsymbol{L}_0, \boldsymbol{S}_0)\|_F$ which depends linearly on the parameter $\delta$.
- The intimate connection between $\sqrt{\text{PCP}}$ formulation (1.2) and StablePCP$_c$ formulation (1.5) can help translate the above solution property to $\sqrt{\text{PCP}}$ (Lemma 2.2).
- The powerful dual certificate construction proposed in [6] (restated in Lemma 2.3) can be used as an approximate subgradient to bound the regularizer at $\widehat{\boldsymbol{X}}$.

The proof of the main theorem has two steps. First, it uses the optimality condition and the subgradient to provide an upper and an lower bound for the regularization terms at $\widehat{\boldsymbol{X}}$. Second, the result in Theorem 2.1 is translated into the $\sqrt{\text{PCP}}$ setting, and together with the bounds obtained above, we get the desired result. The proof is given in the supplementary material, and below we provide three ingredients.

First, we state the main theorem for StablePCP$_c$ problem:

**Theorem 2.1 (Theorem 2 in [9])** *Under Assumptions 1.1 and 1.2, assuming further that $r \leq \rho'_r n_2 \nu^{-1} (\log n_1)^{-2}$ and $m \leq \rho'_s n_1 n_2$ where $\rho'_r, \rho'_s$ are some positive constants, there is a numerical constant $c'$ such that with probability at least $1 - c' n_1^{-10}$, for any $\boldsymbol{Z}_0$ with $\|\boldsymbol{Z}_0\|_F \leq \delta$, the solution $\widehat{\boldsymbol{X}} = (\widehat{\boldsymbol{L}}, \widehat{\boldsymbol{S}})$ to the StablePCP$_c$ problem 1.5 with $\lambda = 1/\sqrt{n_1}$ satisfies*

$$\|\widehat{\boldsymbol{X}} - \boldsymbol{X}_0\|_F \leq \sqrt{320 n_1 n_2 + 4} \cdot \delta. \tag{2.1}$$

Note that choosing $\delta = \|\boldsymbol{Z}_0\|_F$ allows a reconstruction error that is $O(\sqrt{n_1 n_2} \|\boldsymbol{Z}_0\|_F)$. In the case when $\boldsymbol{Z}_0 = \boldsymbol{0}$, StablePCP$_c$ recovers the matrices exactly: $\widehat{\boldsymbol{X}} = \boldsymbol{X}_0$. This is in agreement with the result in [6]. The next lemma connects the two formulations $\sqrt{\text{PCP}}$ and StablePCP$_c$ and the proof is provided in the supplementary material:

**Lemma 2.2** *Consider the $\sqrt{\text{PCP}}$ problem parameterized by $\mu$ and denote the result as $\widehat{\boldsymbol{L}}_{\text{root}}(\mu), \widehat{\boldsymbol{S}}_{\text{root}}(\mu)$, as well as the StablePCP$_c$ formulation parameterized by $\delta$ and denote the result as $\widehat{\boldsymbol{L}}_{\text{stable}}(\delta), \widehat{\boldsymbol{S}}_{\text{stable}}(\delta)$. Define $\delta(\mu) = \|\boldsymbol{D} - \widehat{\boldsymbol{L}}_{\text{root}}(\mu) - \widehat{\boldsymbol{S}}_{\text{root}}(\mu)\|_F$, then*

$$\widehat{\boldsymbol{L}}_{\text{stable}}(\delta(\mu)), \widehat{\boldsymbol{S}}_{\text{stable}}(\delta(\mu)) = \widehat{\boldsymbol{L}}_{\text{root}}(\mu), \widehat{\boldsymbol{S}}_{\text{root}}(\mu). \tag{2.2}$$

Lastly, we show an adapted dual certificate construction:

**Lemma 2.3 (Adapted from [6])** *Under Assumptions 1.1 and 1.2, assume that $r, m$ satisfies*

$$r \leq \rho_r n_2 \nu^{-1} (\log n_1)^{-2}, \quad m \leq \rho_s n_1 n_2, \tag{2.3}$$

*where $\rho_r \leq 1/10, \rho_s$ are some positive constants. Then there is a numerical constant $c$ such that with probability at least $1 - c n_1^{-10}$, there exists $\boldsymbol{W}, \boldsymbol{F}, \boldsymbol{H}$ such that*

$$\boldsymbol{U}\boldsymbol{V}^* + \boldsymbol{W} = \lambda(\text{sign}(\boldsymbol{S}_0) + \boldsymbol{F} + P_\Omega \boldsymbol{H}), \tag{2.4}$$

*where $\boldsymbol{W} \in T^\perp, \|\boldsymbol{W}\| \leq \frac{1}{2}, P_\Omega \boldsymbol{F} = \boldsymbol{0}, \|\boldsymbol{F}\|_\infty \leq \frac{1}{2}$, and $\|P_\Omega \boldsymbol{H}\|_F \leq \frac{1}{260\sqrt{2}}$.*

The dual construction in [6] satisfies $\|P_\Omega \boldsymbol{H}\|_F \le \frac{1}{4}$. However, the proof for Lemma 2.8(b) indicates that $\|P_\Omega \boldsymbol{H}\|_F \le \frac{\sqrt{r}}{n_1^2} \le \frac{\sqrt{r/n_2}}{n_1^{1.5}}$ and we only need to make sure that $\frac{\sqrt{r/n_2}}{n_1^{1.5}} \le \frac{1}{260\sqrt{2}}$. This is a very mild condition, especially when it comes to the high dimensional real data (such as video). If we require that $r \le n_2/10$, then problems of reasonably large dimension suffice, say, $n_1 \ge 120$. And in the extreme case, we can set $\rho_r \le 1/(260\sqrt{2})^2$.

## 3  Solving $\sqrt{\text{PCP}}$ with ADMM

Different from [9] where StablePCP$_u$ (1.6) is solved via *Accelerated Proximal Gradient* method, we solve $\sqrt{\text{PCP}}$ (and StablePCP$_u$) via ADMM-splitting since the objective is not differentiable. To avoid multi-block ADMM which is not guaranteed to converge [29], we define variables $\boldsymbol{X}_1^* = (\boldsymbol{L}_1^*, \boldsymbol{S}_1^*, \boldsymbol{Z}^*)$, $\boldsymbol{X}_2^* = (\boldsymbol{L}_2^*, \boldsymbol{S}_2^*)$, and reformulate problem (1.2) as:

$$\min_{\boldsymbol{X}_1, \boldsymbol{X}_2} \ f(\boldsymbol{X}_1) := \|\boldsymbol{L}_1\|_* + \lambda\|\boldsymbol{S}_1\|_1 + \mu\|\boldsymbol{Z}\|_F \tag{3.1}$$

$$\text{s.t.} \quad \boldsymbol{X}_1 + \begin{bmatrix} -\boldsymbol{I} & \boldsymbol{0} \\ \boldsymbol{0} & -\boldsymbol{I} \\ \boldsymbol{I} & \boldsymbol{I} \end{bmatrix} \boldsymbol{X}_2 = \begin{bmatrix} \boldsymbol{0} \\ \boldsymbol{0} \\ \boldsymbol{D} \end{bmatrix}.$$

The problem (3.1) can be separated into 2 blocks nicely ($\boldsymbol{X}_1$ and $\boldsymbol{X}_2$), which guarantees convergence of ADMM (under additional mild conditions)[12].

Define dual variables $\boldsymbol{Y}^* = (\boldsymbol{Y}_1^*, \boldsymbol{Y}_2^*, \boldsymbol{Y}_3^*)$, the Lagrangian can be written as

$$\mathcal{L}_\rho(\boldsymbol{X}_1, \boldsymbol{X}_2, \boldsymbol{Y}) = \|\boldsymbol{L}_1\|_* + \lambda\|\boldsymbol{S}_1\|_1 + \mu\|\boldsymbol{Z}\|_F + \langle \boldsymbol{L}_1 - \boldsymbol{L}_2, \boldsymbol{Y}_1 \rangle + \frac{\rho}{2}\|\boldsymbol{L}_1 - \boldsymbol{L}_2\|_F^2 + \langle \boldsymbol{S}_1 - \boldsymbol{S}_2, \boldsymbol{Y}_2 \rangle$$

$$+ \frac{\rho}{2}\|\boldsymbol{S}_1 - \boldsymbol{S}_2\|_F^2 + \langle \boldsymbol{L}_2 + \boldsymbol{S}_2 + \boldsymbol{Z} - \boldsymbol{D}, \boldsymbol{Y}_3 \rangle + \frac{\rho}{2}\|\boldsymbol{L}_2 + \boldsymbol{S}_2 + \boldsymbol{Z} - \boldsymbol{D}\|_F^2.$$

We present the update rules[4] as well as the stopping criteria adapted from [12] in Algorithm 1 and `helper()` function in the supplementary material. The stopping criteria takes into account the primal and the dual feasibility conditions, and the algorithm stops when the tolerances set using an absolute and relative criterion are reached.

If we modify the update of $\boldsymbol{Z}$ in Algorithm 1 to $\boldsymbol{Z} \leftarrow \left(\boldsymbol{D} - \boldsymbol{L}_2 - \boldsymbol{S}_2 - \frac{1}{\rho}\boldsymbol{Y}_3\right)/(1 + \mu/\rho)$, we get ADMM for StablePCP$_u$ (1.6).

---

**Algorithm 1** Algorithm for $\sqrt{\text{PCP}}$

---

**Input:** $\boldsymbol{D} \in \mathbb{R}^{n_1 \times n_2}, \lambda, \mu$.
**Output:** $\boldsymbol{L}, \boldsymbol{S} \in \mathbb{R}^{n_1 \times n_2}$.

---

    # *Tolerance levels, max iterations*
    $\epsilon_{\text{abs}} \leftarrow 10^{-6}, \epsilon_{\text{rel}} \leftarrow 10^{-6}, N \leftarrow 5000$
    # *Initialization*
    $\boldsymbol{L}_1, \boldsymbol{L}_2, \boldsymbol{S}_1, \boldsymbol{S}_2, \boldsymbol{Z}, \boldsymbol{Y}_1, \boldsymbol{Y}_2, \boldsymbol{Y}_3 \leftarrow \boldsymbol{0}_{n_1 \times n_2}$
    $\rho \leftarrow 0.1$
    **for** $i = 1, i \le N, i++$ **do**
        # *Save old values temporarily*
        $(\boldsymbol{L}_2', \boldsymbol{S}_2') \leftarrow (\boldsymbol{L}_2, \boldsymbol{S}_2)$
        # *ADMM updates*
        $\boldsymbol{L}_1 \leftarrow \text{prox}_{\frac{1}{\rho}\|\cdot\|_*}\left(\boldsymbol{L}_2 - \frac{1}{\rho}\boldsymbol{Y}_1\right)$
        $\boldsymbol{S}_1 \leftarrow \text{prox}_{\frac{\lambda}{\rho}\|\cdot\|_1}\left(\boldsymbol{S}_2 - \frac{1}{\rho}\boldsymbol{Y}_2\right)$

    $\boldsymbol{Z} \leftarrow \text{prox}_{\frac{\mu}{\rho}\|\cdot\|_F}\left(\boldsymbol{D} - \boldsymbol{L}_2 - \boldsymbol{S}_2 - \frac{1}{\rho}\boldsymbol{Y}_3\right)$
    $\boldsymbol{L}_2 \leftarrow \frac{\left(\boldsymbol{D} - \boldsymbol{Z} + 2\boldsymbol{L}_1 - \boldsymbol{S}_1 + \frac{1}{\rho}(2\boldsymbol{Y}_1 - \boldsymbol{Y}_2 - \boldsymbol{Y}_3)\right)}{3}$
    $\boldsymbol{S}_2 \leftarrow \frac{\left(\boldsymbol{D} - \boldsymbol{Z} + 2\boldsymbol{S}_1 - \boldsymbol{L}_1 + \frac{1}{\rho}(2\boldsymbol{Y}_2 - \boldsymbol{Y}_1 - \boldsymbol{Y}_3)\right)}{3}$
    $\boldsymbol{Y}_1 \leftarrow \boldsymbol{Y}_1 + \rho(\boldsymbol{L}_1 - \boldsymbol{L}_2)$
    $\boldsymbol{Y}_2 \leftarrow \boldsymbol{Y}_2 + \rho(\boldsymbol{S}_1 - \boldsymbol{S}_2)$
    $\boldsymbol{Y}_3 \leftarrow \boldsymbol{Y}_3 + \rho(\boldsymbol{L}_2 + \boldsymbol{S}_2 + \boldsymbol{Z} - \boldsymbol{D})$
    # *Update $\rho$ and check convergence*
    $\rho, \text{ifConverge} \leftarrow$ `helper()`
    **if** ifConverge **then**
        **break**
    **end if**
  **end for**
  $(\boldsymbol{L}, \boldsymbol{S}) \leftarrow ((\boldsymbol{L}_1 + \boldsymbol{L}_2)/2, (\boldsymbol{S}_1 + \boldsymbol{S}_2)/2)$
  **return** $\boldsymbol{L}, \boldsymbol{S}$

---

[4]Recall that $\text{prox}_{\gamma\|\cdot\|_*}(\boldsymbol{Z}) = \sum_i \max(\lambda_i - \gamma, 0)\boldsymbol{u}_i\boldsymbol{v}_i^*$, where $\boldsymbol{Z} = \sum_i \lambda_i\boldsymbol{u}_i\boldsymbol{v}_i^*$ is the SVD, $[\text{prox}_{\gamma\|\cdot\|_1}(\boldsymbol{Z})]_{i,j} = \max(|\boldsymbol{Z}_{i,j}| - \gamma, 0) \cdot \text{sign}(\boldsymbol{Z}_{i,j})$, and $\text{prox}_{\gamma\|\cdot\|_F}(\boldsymbol{Z}) = \max(\|\boldsymbol{Z}\|_F - \gamma, 0)\frac{\boldsymbol{Z}}{\|\boldsymbol{Z}\|_F}$.

## 4  Experiments

To show the effectiveness of our new formulation, we test $\sqrt{\text{PCP}}$ on simulated data as well as real-world video datasets. The experiments suggest that our error bound in Theorem 1.1 has a correct dependency on the noise level of $\boldsymbol{Z}_0$, but loses a factor of $n$ (the dimension of the problem). In addition, the solutions produced by $\sqrt{\text{PCP}}$ with our proposed noise-independent $\mu$ and StablePCP$_u$ with the noise-dependent $\mu$ often look very similar to each other. Moreover, experiments on real-world datasets with natural noise also show the denoising effect of $\sqrt{\text{PCP}}$, making $\sqrt{\text{PCP}}$ a practical approach with good performance in this robust noisy low-rank matrix recovery setting.

Additional experiments of $\sqrt{\text{PCP}}$ on simulated data with varying $\mu$ also suggest that $\sqrt{n_2/2}$ can provide performance (recovery error) close to the optimal $\mu$, justifying our proposed choice of $\mu = \sqrt{n_2/2}$.

### 4.1  Simulations with Varying Noise Levels and Dimension

In this set of experiments, we are interested in how our error bound in Theorem 1.1 compare with the actual reconstruction error. We simulate $(\boldsymbol{L}_0, \boldsymbol{S}_0, \boldsymbol{Z}_0)$ with varying noise levels of $\boldsymbol{Z}_0$ and problem dimension $n_1, n_2$. To simulate $\boldsymbol{L}_0 \in \mathbb{R}^{n_1 \times n_2}$ of rank $r$, we generate $\boldsymbol{U} \in \mathbb{R}^{n_1 \times r}, \boldsymbol{V} \in \mathbb{R}^{n_2 \times r}$ as the unnormalized singular vectors such that $\boldsymbol{U}, \boldsymbol{V}$ are entrywise i.i.d. $\mathcal{N}(0, 1/n_1)$ and $\mathcal{N}(0, 1/n_2)$ respectively and let $\boldsymbol{L}_0 = \boldsymbol{U}\boldsymbol{V}^*$. For $\boldsymbol{S}_0$, we let $P[(i,j) \in \Omega] = \rho_S$ and for $(i,j)$ in support $\Omega$, $(\boldsymbol{S}_0)_{(i,j)} \in \{0.05, -0.05\}$ with equal probability. For the noise $\boldsymbol{Z}_0$, we generate it as entrywise i.i.d. $\mathcal{N}(0, \sigma^2)$.

In addition, in the experiments we take $n_1 = n_2 = n$, so we choose $\lambda = 1/\sqrt{n}$, $\mu_{\text{stable}} = 1/(2\sigma\sqrt{n})$ (the noise level $\sigma$ is known), and $\mu_{\text{root}} = \sqrt{n/2}$. Theoretical analysis in Theorem 1.1 and 2.1 shows that with these parameters, $\|\widehat{\boldsymbol{X}} - \boldsymbol{X}_0\|_F = O(n\|\boldsymbol{Z}_0\|_F)$.

To test the dependency of the error on $\sigma$, we take $n = 200$, $r = 10$, and so $\|\boldsymbol{L}_0\|_F^2 \approx r = 10$. For the outlier $\boldsymbol{S}_0$, we take $\rho_S = 0.1$, so $\|\boldsymbol{S}_0\|_F^2 \approx 0.05^2 n^2 \rho_S = 10$. For the noise $\boldsymbol{Z}_0$, we take $\sigma \in \{0, 0.001, \ldots, 0.015\}^5$, so $\|\boldsymbol{Z}_0\|_F^2 \approx \sigma^2 n^2 \in [0, 9]$. For each $\sigma$ in the given set, we randomly generate 20 ground truth $(\boldsymbol{L}_0, \boldsymbol{S}_0, \boldsymbol{Z}_0)$ triplets and run $\sqrt{\text{PCP}}$ and StablePCP$_u$ on them. We use the root-mean-squared (RMS) error defined as $(\frac{1}{20} \sum_{k=1}^{20} \|\widehat{\boldsymbol{L}}^{(k)} - \boldsymbol{L}_0\|_F^2)^{1/2}$ and $(\frac{1}{20} \sum_{k=1}^{20} \|\widehat{\boldsymbol{S}}^{(k)} - \boldsymbol{S}_0\|_F^2)^{1/2}$ for evaluation. In Figure 1(a) we show the RMS error over 20 trials for the low-rank and the sparse. It is clear from the plot that $\|\widehat{\boldsymbol{L}} - \boldsymbol{L}_0\|_F$ and $\|\widehat{\boldsymbol{S}} - \boldsymbol{S}_0\|_F$ are $O(\sigma)$ for both $\sqrt{\text{PCP}}$ and StablePCP$_u$, which confirms that the reconstruction error is linear in the noise level $\sigma$.

We also notice that the recovery error in Figures 1(a) and 1(c) is linear in the noise level for small $\sigma$, but exhibits a sublinear behavior for larger $\sigma$. This behavior reflects a general phenomenon in recovery/denoising using structured models (sparse, low-rank, etc.): the minimax noise sensitivity $\eta = \sup_{\sigma>0} \frac{1}{\sigma} E[\|\widehat{\boldsymbol{x}} - \boldsymbol{x}_0\|]$ is obtained as $\sigma \to 0$. This means that for small $\sigma$, we expect a linear trend with slope $\eta$, while for larger $\sigma$, the dependence can be sublinear. This behavior has a general geometric explanation. For simplicity we sketch how this plays out in a simpler norm denoising problem, in which the target is to recover a structured signal $\boldsymbol{x}_0$, and we observe $\boldsymbol{y} = \boldsymbol{x}_0 + \sigma\boldsymbol{z}$. For simplicity, assume that we know that $\|\boldsymbol{x}_0\|_1 \leq \tau$, and solve $\min_{\|\boldsymbol{x}\|_1 \leq \tau} \|\boldsymbol{x} - \boldsymbol{y}\|_2$. For small $\sigma$, the estimation error $\widehat{\boldsymbol{x}} - \boldsymbol{x}_0$ is simply the projection of the noise $\sigma\boldsymbol{z}$ onto the descent cone of the norm ball $\{\|\boldsymbol{x}\|_1 \leq \tau\}$ at $\boldsymbol{x}_0$; its size is linear in $\sigma$. For larger $\sigma$, there is additional denoising due to the fact that the L1 ball is smaller than the descent cone at $\boldsymbol{x}_0$ — this leads to the behavior observed here.

To test the dependency of the error on the problem dimension, we vary $n \in \{200, 300, \ldots, 1000\}$ and take $r = 0.1n$. We keep the setting for $\boldsymbol{S}_0$, and take $\sigma = 0.01$ as the noise level for $\boldsymbol{Z}_0$. Figure 1(b) shows the RMS error. Note that for fixed $\sigma$, $\|\boldsymbol{Z}_0\|_F \sim n\sigma$, so the results in Theorem 1.1 and Theorem 2.1 bound the reconstruction error as $O(n^2)$.

---

[5]When $\sigma = 0$ and $\mu_{\text{stable}} = +\infty$, StablePCP$_u$ is equivalent to StablePCP$_c$ with $\delta = 0$.

However, the analysis provides only a loose error bound. As can be seen from this set of experiment, the error is closer to $O(n)$. We provide experiments with different distributions of the noise in the appendix.

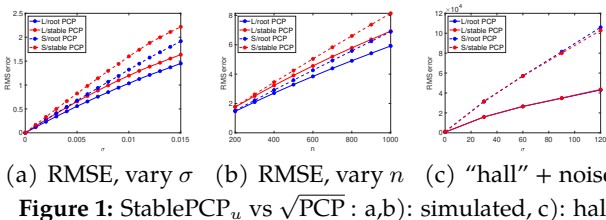

(a) RMSE, vary $\sigma$    (b) RMSE, vary $n$    (c) "hall" + noise

**Figure 1:** StablePCP$_u$ vs $\sqrt{\text{PCP}}$ : a,b): simulated, c): hall

## 4.2 Real Data with Added Noise: Surveillance Video

Many imaging datasets can be modeled as the sum of a low-rank matrix $\boldsymbol{L}_0$, a sparse outlier $\boldsymbol{S}_0$, and noise $\boldsymbol{Z}_0$. For instance, video data often consists of an almost fixed background which can be seen as low-rank, and a foreground (such as people) that only occupies a small fraction of the image pixels for a short amount of time, which can be considered as sparse. Thus, videos can naturally fit into our robust PCA framework.

In this set of experiments, we use $\sqrt{\text{PCP}}$ and StablePCP$_u$ to separate the background and the foreground for surveillance video data. We assume that the original video is noiseless, and manually add noise $\boldsymbol{Z}_0$ that is entrywise i.i.d. $\mathcal{N}(0, \sigma^2)$ to test the dependency of the reconstruction error on the noise level $\sigma$.

We use the "hall dataset" in [30], a 200-frame video of a hall that has people walking around. Each frame has resolution $144 \times 176$, and is flattened as one column of the noiseless observation matrix $\boldsymbol{D}$, so we have $n_1 = 144 \times 176$ and $n_2 = 200$. Each pixel is represented by a number in $[0, 255]$, and the mean value among all pixels is $150.3295$, with standard deviation $45.7438$, and median $155.0000$.

For the added noise, we choose $\sigma \in \{0, 30, 60, 90, 120\}$, and denote the recovered matrices as $\widehat{\boldsymbol{X}}^{(\sigma)}_{\text{root/stable}}$. In addition, we let $\boldsymbol{X}_0 = \frac{1}{2}(\widehat{\boldsymbol{X}}^{(0)}_{\text{root}} + \widehat{\boldsymbol{X}}^{(0)}_{\text{stable}})$ be the ground truth, and evaluate the error using $\|\widehat{\boldsymbol{L}}^{(\sigma)}_{\text{root/stable}} - \boldsymbol{L}_0\|_F$ and $\|\widehat{\boldsymbol{S}}^{(\sigma)}_{\text{root/stable}} - \boldsymbol{S}_0\|_F$. We take $\lambda = 1/\sqrt{n_1}$, $\mu_{\text{root}} = \sqrt{n_2/2}$ and $\mu_{\text{stable}} = \frac{1}{\sigma(\sqrt{n_1}+\sqrt{n_2})}$ following the same intuition as in Section 1.[6] We run the experiments on a laptop with 2.3 GHz Dual-Core Intel Core i5, and set the maximal iteration of ADMM to be 5000. All of these experiments on real datasets end within 1 hour. For full details, please see the supplementary material.

In Figure 1(c), we show the reconstruction error with varying noise levels. It can be seen that the error is indeed linear in $\sigma$, as predicted by our analysis. In Figures 2, we present the first frame (i.e. the first column) of the original video (with noise $\sigma = 0, 30$), and the $\sqrt{\text{PCP}}$ recovered low-rank and sparse matrices. Although the added noise blurs the videos, our $\sqrt{\text{PCP}}$ is still stable and successfully decompose the background and the foreground.

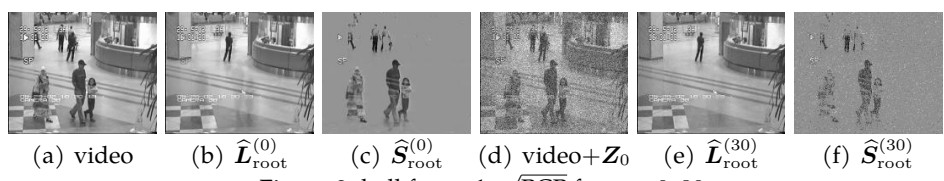

(a) video    (b) $\widehat{\boldsymbol{L}}^{(0)}_{\text{root}}$    (c) $\widehat{\boldsymbol{S}}^{(0)}_{\text{root}}$    (d) video+$\boldsymbol{Z}_0$    (e) $\widehat{\boldsymbol{L}}^{(30)}_{\text{root}}$    (f) $\widehat{\boldsymbol{S}}^{(30)}_{\text{root}}$

**Figure 2:** hall frame 1: $\sqrt{\text{PCP}}$ for $\sigma = 0, 30$

## 4.3 Real Data with Natural Noise: Low Light Video

Low light videos are known to have very large observation noise due to limited photon counts. In this experiment, we apply our $\sqrt{\text{PCP}}$ to the Dark Raw Video (DRV) dataset in [32] (under MIT License) for foreground background separation and denoising. This dataset of

---

[6]Recall that $E[\|\boldsymbol{Z}_0\|] \leq \sigma(\sqrt{n_1} + \sqrt{n_2})$ for rectangular matrices, e.g. from [31]

RGB videos, approximately 110 frames each, $3672 \times 5496$ in resolution, was collected at low light settings, so the signal-to-noise ratio (SNR) is extremely low (negative if measured in dB)[32].

For the experiments, we choose video M0001 (basketball player), M0004 (toy windmill), and M0009 (billiard table) from DRV. As preprocessing, we convert the RGB videos to grayscale using `rgb2gray()` in Matlab, and crop and downsample each frame to reduce data size. The final resolution is $322 \times 440$ for M0001, $294 \times 440$ for M0004, and $306 \times 458$ for M0009.

We apply $\sqrt{\text{PCP}}$ with $\lambda = 1/\sqrt{n_1}$, and $\mu = \sqrt{n_2/2}$ to these 3 videos, and present the results for frame 30 in Figure 3. The denoising effect of $\sqrt{\text{PCP}}$ can be seen by comparing $\boldsymbol{D}$ with $\widehat{\boldsymbol{L}}+\widehat{\boldsymbol{S}}$. In addition, $\widehat{\boldsymbol{L}}$ recovers the background pretty well, $\widehat{\boldsymbol{S}}$ captures the moving foreground but is still mixed with noise, which we believe is due to the extremely low SNR.

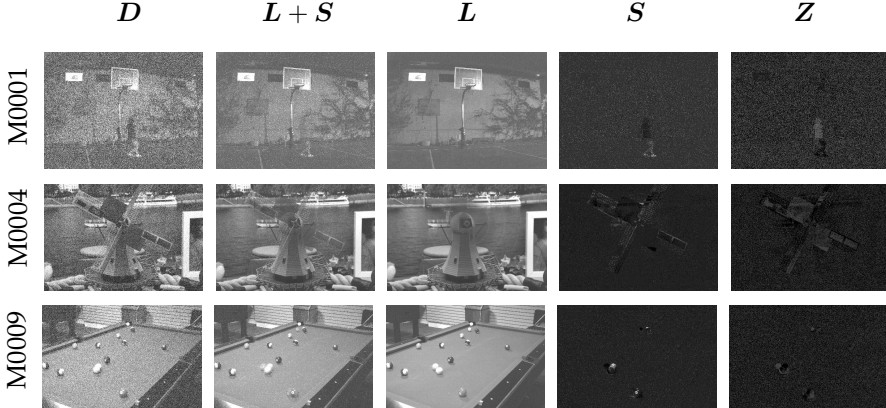

**Figure 3:** Low light video frame 30 for M0001, M0004, and M0009 ($\widehat{\boldsymbol{Z}} = \widehat{\boldsymbol{L}} + \widehat{\boldsymbol{S}} - \widehat{\boldsymbol{D}}$). Image contrast is enhanced using `imadjustn()` in Matlab.

### 4.4 Real Data with Natural Noise: Optical Coherence Tomography

In medical imaging, Optical Coherence Tomography can be used for micro-scale resolution, quick scanning of biological phenomenon [33]. These scans of the same scene over time, called time-lapse B-scan, are often noisy, but fit into our low rank/sparse model.

In this experiment, we apply $\sqrt{\text{PCP}}$ to the time-lapse B-scans (250 frames of resolution $300 \times 150$) of human trachea samples containing motile cilia (demo dataset of [33] under CC0 License). We present the recovered frame 50 and 100 in Figure 4. As expected, $\widehat{\boldsymbol{L}}$ captures the static background, and $\widehat{\boldsymbol{S}}$ captures the motion of cilia.

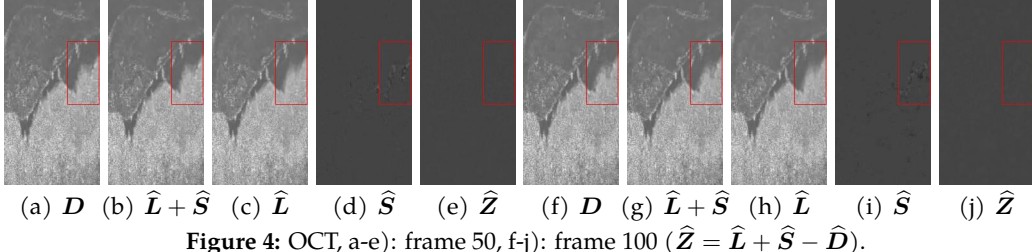

(a) $\boldsymbol{D}$ (b) $\widehat{\boldsymbol{L}}+\widehat{\boldsymbol{S}}$ (c) $\widehat{\boldsymbol{L}}$ (d) $\widehat{\boldsymbol{S}}$ (e) $\widehat{\boldsymbol{Z}}$ (f) $\boldsymbol{D}$ (g) $\widehat{\boldsymbol{L}}+\widehat{\boldsymbol{S}}$ (h) $\widehat{\boldsymbol{L}}$ (i) $\widehat{\boldsymbol{S}}$ (j) $\widehat{\boldsymbol{Z}}$

**Figure 4:** OCT, a-e): frame 50, f-j): frame 100 ($\widehat{\boldsymbol{Z}} = \widehat{\boldsymbol{L}} + \widehat{\boldsymbol{S}} - \widehat{\boldsymbol{D}}$).

### 4.5 Optimal Choice of $\mu$

Our main result Theorem 1.1 suggests a tuning-free $\mu = \sqrt{n_2/2}$. Here, we investigate experimentally if this choice of $\mu$ is optimal. We vary the problem dimensions $n_1$ and $n_2$, the rank-dimension ratio $\rho_L := r/n$ ($n = n_1 = n_2$), and the noise standard deviation $\sigma$. For each choice of parameters $(n_1, n_2, \rho_L, \sigma)$, we generate 10 pairs of $(\boldsymbol{L}_0, \boldsymbol{S}_0, \boldsymbol{Z}_0)$ using the same method as in Section 4.1, run $\sqrt{\text{PCP}}$ with $\lambda = 1/\sqrt{n_1}$ and $\mu = c\mu_0$, where $\mu_0 = \sqrt{n_2}$ and $c$ is a varying coefficient ($c = 1/\sqrt{2} \approx 0.71$ corresponds to our proposed value). Settings of all parameters are included in the supplementary material.

We use the 10-average of $\|(\widehat{L}, \widehat{S}) - (L_0, S_0)\|_F$ as the evaluation metric. In Figure 5, we show the heatmaps of this metric relative to the optimal $\mu = c\mu_0$ among all tested $c$, i.e. $\eta_{\text{rel}}(\mu) = \frac{\|(\widehat{L}(\mu), \widehat{S}(\mu)) - (L_0, S_0)\|_F}{\min_{\mu' = c\mu_0} \|(\widehat{L}(\mu'), \widehat{S}(\mu')) - (L_0, S_0)\|_F}$, so the optimal $\mu$ has value 1 in each row of the heatmaps. From Figure 5, we see that varying $n_1$, $n_2$ has little effect on the optimal choice

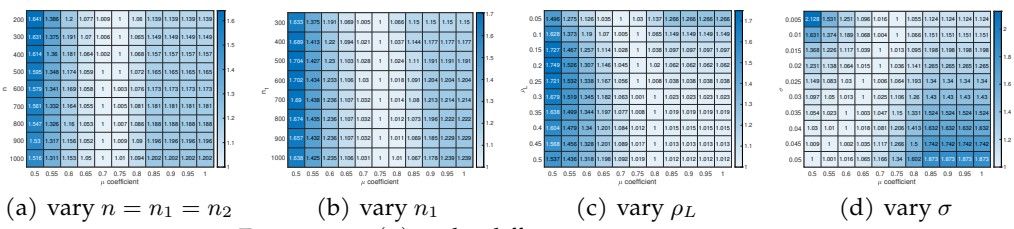

| (a) vary $n = n_1 = n_2$ | (b) vary $n_1$ | (c) vary $\rho_L$ | (d) vary $\sigma$ |

**Figure 5:** $\eta_{\text{rel}}(\mu)$ under different varying parameters

of $\mu$, which is approximately between $0.7\sqrt{n_2}$ and $0.75\sqrt{n_2}$, close to our $\sqrt{n_2/2}$. However, decreasing $\rho_L$ or increasing $\sigma$ suggests a smaller value of optimal $\mu$. This makes sense because with higher level of noise or smaller rank (thus smaller norm $\|L_0\|_F$), the SNR is smaller, so we should put smaller penalty on $\|L + S - D\|_F$. Nevertheless, in all these settings, choosing $\mu = \sqrt{n_2/2}$ still gives satisfying results, as the recovery errors for $\mu = 0.7\sqrt{n_2}$ are close to the optimal performance: the error ratios are below $1.2$. From these results, we believe that while $\mu = \sqrt{n_2/2}$ may not be the optimal choice with respect to the recovery error, it can provide performance close to the optimal, and is therefore a very effective choice.

## 5   Conclusion

In this work, we propose $\sqrt{\text{PCP}}$, a convex optimization approach for noisy robust low-rank matrix recovery. The benefit of our approach as compared to previous methods such as stable PCP is that it enables tuning-free recovery of low-rank matrices: theoretical analysis and simulations show that a single universal penalty parameter yields stable recovery at any noise standard deviation. Real video data experiments show suggest that many real life models fit into this low-rank plus sparse setting, and $\sqrt{\text{PCP}}$ (as well as stable PCP) does a good job in denosing and recovering the patterns of interest.

The presented experiments suggest the potential for both positive and negative societal impacts: visual surveillance can be abused, leading to significant negative impacts; at the same time, the denoising and foreground/background separation ability of $\sqrt{\text{PCP}}$ can help improve the quality of noisy data in biomedical and scientific research (e.g. medical imaging), and people's life (e.g. low light video).

## Acknowledgement

We would like to thank Marianthi-Anna Kioumourtzoglou, Jeff Goldsmith, Elizabeth Gibson, Rachel Tao and Lawrence Chillrud for many helpful discussions around matrix modeling of environmental data and the need for tuning-free solutions. This work was partially funded by the National Institute of Environmental Health Sciences (NIEHS) grant R01 ES028805. We also thank Christine Hendon for helpful pointers regarding optical coherence tomography data. Jingkai Yan also gratefully acknowledges support from the Wei Family Foundation.

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
