# OpenReview forum: "Square Root Principal Component Pursuit: Tuning-Free Noisy Robust Matrix Recovery"
_NeurIPS.cc/2021/Conference — NeurIPS 2021 Poster_

### Official Review · Reviewer_kjYV · 2021-07-16

**Rating:** 6
**Confidence:** 4

**Summary:**

Motivated from square-root Lasso, this paper propose a new tuning-free convex program, called SR-PCP, to solve low-rank matrix recovery problem from observations corrupted by sparse outliers and dense noise.

**Limitations And Societal Impact:**

This is a theoretical work, and I think there is no potential societal impact.

**Main Review:**

Strengths:
1. This paper proposes a new program which has appealing tuning-free property, and also provide theoretical guarantees as well as an ADMM algorithm to solve it. The theory is solid, and the paper is clearly written.
1. SR-PCP improves the prior work [8] in the sense that one does not need to tune the regularization parameter, which either requires to estimate the noise level (which might be difficult) or using cross-validation (which lacks theoretical guarantee). Although the idea is not new (which originates from square-root lasso [11]), but it is still meaningful to introduce this idea to the field of low-rank matrix recovery.
2. The authors proved stability bounds SR-PCP which scales with the noise level. This stability bound guarantees exact recovery when there is no noise, even in the presence of outliers.

Weakness:
The main weakness of this paper, as the authors mentioned in Section 1.3, is that their estimation error bound is sub-optimal by a dimension-dependent factor. I think it would be better if the authors can provide more explanation on why this sub-optimality occurs compared to prior works, and if it is possible to generlize the proof of this paper to achieve the optimal rates.

**Time Spent Reviewing:**

1.5

---

> ### Author Response · Authors · 2021-08-11
> **Response to Reviewer kjYV**
>
> Thank you for the valuable feedback! Regarding the weakness, we will add more discussion about the sub-optimal error bound in the revised version.
>
> We believe that this sub-optimality mainly comes from the stable PCP work (Theorem 2.1 as cited). In the proof of our main theorem, the final step combines (A4) and (A5). If Theorem 2.1/(A5) can be improved from $O(n\delta)$ to $O(\delta)$, together with (A4), an $O(\lVert \boldsymbol Z_0 \rVert_F)$ bound can be obtained.

---

### Official Review · Reviewer_7yYW · 2021-07-17

**Rating:** 5
**Confidence:** 4

**Summary:**

Low-rank matrix recovery in the presence of sparse corruptions and random noise, under incoherence assumptions on the low-rank and on the sparse components, have been considered. Following an existing line of work, a tuning-free optimization programs has been proposed for demixing and denoising. The analysis, utilizing existing techniques, establishes a bound on the reconstruction error. The bound has been observed to be loose by a dimension-dependent factor (according to the experiments and compared to other estimators for this problem). The estimator is a convex optimization problem which authors solve via ADMM.  Numerical experiments support the usefulness of the proposed approach in a set of synthetic and real data scenarios.

**Limitations And Societal Impact:**

The proposal is a natural continuation on the idea behind the square root lasso and a few other tuning-free estimators. While such continuation of the underlying idea is undoubtedly valuable, I think a thorough examination of other aspects (beyond the design and the analysis of the estimator; e.g., tuning, computation, runtime, etc) could add further unique value. I believe the authors have already covered a number of interesting aspects (a bound, an algorithm, and the numerical experiments: varying mu, comparing the solutions themselves, etc). However, some further aspects could enhance the paper. Here are a few suggestions:
- comparing the actual running time of 1) the other two formulations including their hyperparameter tuning 2) with the current proposal considering the quality of solutions (in an appropriae metric) and the possible need for tuning \mu (to improve the quality of the solution); considering the comparable performance reported in Figure 1 in the Appendix, etc. It would be helpful to see such comparison in various data regimes.
- further discussions on the dimension-dependent factor.
- experiments with different noise distributions (which affects the range for \sigma), closer to the distributions in real data (vision or others).



======================== Update

I have read the other reviews as well as the responses. I would like to thank the authors for their clear explanations. I think the proposed additions are satisfactory; except for the explanations in regards to:

1) the sub-optimality of the bound, where I would like to (not necessarily as part of this review process) see further intuitions and arguments beyond the mechanics of the current proof.

2) the presented experiments. I apologize as I can see that my original comment (the first bullet point in the above) is not very clear. What I meant was to suggest examining a possible scenario in which sqrt-PCP becomes overall computationally-expensive due to a requirement for tuning mu (motivation: sqrt-PCP has an additional non-smoothness compared to PCP, and, ADMM is a choice which requires examination). Let me elaborate:

It would be helpful if the authors could use a clear metric(s) for the the quality of the solutions (at the moment, in the Appendix, there are errors reported in Tables 1,2, and there are figures that are not easy to compare quantitatively) and then report wall-clock times for all three algorithms to achieve pre-specified levels of accuracy according to said metric (at the moment, the numbers in Tables 1,2, while comparable between the corresponding cells in the two tables, seem rather arbitrary). Note that in the second step, the desired accuracy determines the length of search for an appropriate mu. Tuning mu, if needed, would increase the wall-clock time for sqrt-PCP, and allows for understanding the limitations of the tuning-free estimator and the solver (e.g., in high-accuracy scenarios). Note that in this experimental setup, tuning for mu is not a choice but is rather dictated by the noise and the desired accuracy level. Finally, considering various data regimes allows for examining the above question when starting the search with the data-agnostic choice of mu=sqrt(n_2/2).

I would like to emphasize that this suggestion is only to 'stress' the system in order to understand its limitations, and I would understand if the authors believe this is outside the scope of this work. However, I believe such examination is important in supporting the tuning-free claim for the proposal (in conjunction with the theoretical bounds). It would be nice to hear the authors' thoughts and whether I am missing a technical aspect in making this suggestion.

**Main Review:**

I believe the work is original (of course, building upon a now well-known idea as in square-root lasso). The manuscript is clear and details have been provided. However, considering existing works on this approach, of design and analysis for a tuning-free regularized estimator, I think further examination is required to establish this proposal as a useful method in practice.

line 59: I am not sure if the use of 'separable' (which I assume points to the splitting in ADMM) is appropriate.

The bullet point starting on line 165 seem to be referring to the proof of Theorem 1.1, but is rather technical and vague without further details. A rephrasing of this item could be helpful.

Proof of Lemma 2.2 in the Appendix has a few broken references.

Appendix, the equations after (A.3), the third inequality:
1) bounding r seems to require \nu\geq 1
2) it seems that 1/520 should be 1/(520 \sqrt{n_1 n_2}), but the fourth inequality is correct.
I am not sure if I understand why A.5 holds. Otherwise, the rest of the proof goes through.

line 245: could you please elaborate on how the plot has been interpreted as O(\sigma)?
Moreover, the plots suggest a concave trend (authors concluded a linear trend; lines 246, 276). Any insights would be appreciated.

The visual inspections in sections 4.3 and 4.4 are difficult; Figures 3 and 4. Alternative presentation of the intended results and conclusions could be helpful.

It might be helpful to discuss the 'limitations' incurred by the homogeneity of the objective function (discussed on lines 106-112); for example in tuning or solving the optimization problem, etc.

Do the authors have any intuition on the dimension-dependent factor in their bound? (lines 132-133, 159)


**Time Spent Reviewing:**

4

---

> ### Author Response · Authors · 2021-08-11
> **Response to Reviewer 7yYW**
>
> Thank you for the very detailed review and all the suggestions! Regarding the comments in the Main Review section,
>
> -	The word "separable" on line 59 indeed refers to "splittable" in ADMM, namely the objective function is the sum of functions of different variables. We will clarify this in the revised version.
> -	What the bullet point on line 165 tries to emphasize is that the stable PCP error bound in Theorem 2.1 plays an important role in the proof or our main theorem, as it bounds the norm of $(\widehat{\boldsymbol\delta_L}, \widehat{\boldsymbol\delta_S})$ by $\delta$, which is related to the norm of $\widehat{\boldsymbol\delta_L} + \widehat{\boldsymbol\delta_S}$. Here we use the notations $\widehat{\boldsymbol\delta_L} = \widehat{\boldsymbol L} - \boldsymbol L_0$ and $\widehat{\boldsymbol\delta_S} = \widehat{\boldsymbol S} - \boldsymbol S_0$ as defined in Appendix A. We agree that this bullet point might be unnecessarily technical, and will simplify it.
> -	Sorry for any inconvenience and misunderstanding due to the broken references! We will fix them!
>
> -	For the details in the proof:
> --	According to the definition of $\nu$ in Assumption 1, $\lVert \boldsymbol U \rVert_F^2 = r$, so the maximum row norm squared is $\geq r/n_1$, and so $\nu\geq 1$, which gives the bound $r\leq n_2/10$.
> --	To get (A5), we first use Lemma 2.2 to connect the solutions to root PCP and stable PCP (with the specified $\delta = \lVert \widehat{\boldsymbol\delta_L} + \widehat{\boldsymbol\delta_S} - \boldsymbol Z_0 \rVert_F)$. Due to this connection, Theorem 2.1 (sorry again for the broken reference) can be applied to bound $\lVert (\widehat{\boldsymbol\delta_L}, \widehat{\boldsymbol\delta_S}) \rVert_F$ by $\delta (= \lVert \widehat{\boldsymbol\delta_L} + \widehat{\boldsymbol\delta_S} - \boldsymbol Z_0 \rVert_F)$.
> --	However, we realize that Theorem 2.1 is only applicable when $\delta \geq \lVert \boldsymbol Z_0 \rVert_F$. Nevertheless, it’s not difficult to extend Theorem 2.1 to unconstrained $\delta$, with the $\delta$ in the bound (2.1) replaced by $(\delta+\lVert \boldsymbol Z_0 \rVert_F)/2$. The constant factor in the error bound in Theorem 1.1 can be improved slightly with the new version of Theorem 2.1. We will fix this, and add more details and comments to help readers understand the proof!
>
> -	Regarding the sublinear trend observed in the graph on line 245: This is an excellent question! As the reviewer correctly notes, the recovery error is linear in the noise level $\sigma$ for small $\sigma$, but exhibits a sublinear behavior for larger $\sigma$.
> --	This behavior reflects a general phenomenon in recovery/denoising using structured models (sparse, low-rank, etc.): the minimax noise sensitivity $\eta = \sup_{\sigma >0} \tfrac{1}{\sigma} \mathbb{E} \| \hat{x} - x_0 \|$ is obtained as $\sigma \to 0$. This means that for small $\sigma$, we expect a linear trend with slope $\eta$, while for larger $\sigma$, the dependence can be sublinear.
> --	This behavior has a general geometric explanation; for simplicity we sketch how this plays out in a simpler norm denoising problem, in which the target is to recover a structured signal $\boldsymbol x_0$, and we observe $\boldsymbol y = \boldsymbol x_0 + \sigma \boldsymbol z$. For simplicity, assume that we know that $\lVert \boldsymbol x_0 \rVert_1 \le \tau$, and solve
> $$\min_{\lVert \boldsymbol x \rVert_1 \le \tau} \lVert  \boldsymbol x - \boldsymbol y \rVert_2.$$
> For small $\sigma$, the estimation error $\hat{\boldsymbol x} - \boldsymbol x_0$ is simply the projection of the noise $\sigma \boldsymbol z$ onto the descent cone of the norm ball $\lbrace \lVert  \boldsymbol x \rVert_1 \le \tau \rbrace$ at $\boldsymbol x_0$; its size is linear in $\sigma$. For larger $\sigma$, there is additional denoising due to the fact that the L1 ball is smaller than the descent cone at $\boldsymbol x_0$ — this leads to the behavior observed here.
> --	In the final version, we will add a discussion of this issue and clarify our comments regarding the linear dependence of the estimation error on the noise level.
>
> -	We agree that Figures 3 and 4 might be challenging to interpret, and this is partly caused by the data collection settings (low light/OCT datasets are naturally noisy and low-contrast). We supplement the figures with enhanced contrast versions, and highlight the key parts, such as the moving cilia in OCT, with red circles.
> -	The square root term makes the objective homogeneous, but the term is no longer smooth or differentiable, and so we cannot apply algorithms such as the proximal gradient method (which requires the objective to be the sum of a smooth and a non-smooth function). This also becomes our incentive to use ADMM. Thanks for pointing this out! We will add notes about this to the revised version.
> -	For the dimension-dependent factor in the error bound, we believe that the sub-optimality mainly comes from the stable PCP work (Theorem 2.1 as cited). In the proof of our main theorem, the final step combines (A4) and (A5). If Theorem 2.1/(A5) can be improved from $O(n\delta)$ to $O(\delta)$, together with (A4), an $O(\lVert \boldsymbol Z_0 \rVert_F)$ bound can be obtained.
>
> Thank you for the valuable suggestions in the Limitations and Societal Impact section!
>
> -	Regarding running time comparisons for hyperparameter search: We thank the reviewer for this suggestion. Indeed, the major advantage of $\sqrt{\text{PCP}}$ is that it allows one to set the regularization parameter without needing to know the noise level in advance, making it easier to tune parameters. As the reviewer correctly suggests, if one sets out to tune the $\mu$ parameter (say by some kind of cross-validation strategy), this should result in significantly reduced wall clock time, because one would need to solve fewer instances of $\sqrt{\text{PCP}}$. We agree with this intuition — and indeed, compared to PCP our ADMM implementation of $\sqrt{\text{PCP}}$ requires less wall clock time to solve a single instance (see Section C.1 of the supplemental material).
> -	The main challenge in extending these results to a controlled experiment on parameter search is the number of degrees of freedom: optimization hyperparameters and stopping criteria for both methods, the search grid and search strategy, etc. We will add comments about this issue in the final version, and defer a detailed investigation to future work.
> -	As mentioned above, we will add more discussions on the dimension-dependent factor.
> -	Regarding other noise distributions, this is a good suggestion! We will add experiments in which the noise (entries of $\boldsymbol Z_0$) follows Poisson or uniform distributions, which are common in vision datasets. We believe that our root PCP is still robust to such noise, since the main result Theorem 1.1 is independent of the noise distribution.

---

### Official Review · Reviewer_WLTB · 2021-07-17

**Rating:** 7
**Confidence:** 4

**Summary:**

The paper describes a square-root principal component pursuit for low rank matrix recovery, which is a modification of the original PCP approach in which the original squared Frobenius norm term is replaced with a non-squared term. The paper shows that this change allows one to set the weights of the regularizer in a manner independent of the noise level.

**Limitations And Societal Impact:**

The paper states the noise regimes in which other algorithms exhibit better noise tolerance than the proposed approach; nonetheless, those must be aware of the noise level. Application-dependent social impacts are discussed.

**Main Review:**

The notation (L,S) should be defined explicitly - is it a concatenation? It appears inside a Frobenius norm in line 169.

In Lemma 2.3, it is not clear what T, U and V are. Perhaps one should write L0 instead of UV*?

In Algorithm 1, although the proximal operators for ell1 and nuclear norm are well know, it is less clear what the proximal operator for the (non squared) Frobenius norm is. Is it a scaling of the matrix by its Frobenius norm? Adding detail on these would be welcome.

It would be good to discuss why the results in Figure 1(c) show no improvement for root PCP. It would also be good to have numerical comparisons with PCP with real data, perhaps in the supplemental material.

**Time Spent Reviewing:**

5 hours

---

> ### Author Response · Authors · 2021-08-10
> **Response to Reviewer WLTB**
>
> Thank you for the thorough review and for pointing out these issues! Regarding notations and clarity,
> -	The notation $(\boldsymbol L, \boldsymbol S)$ represents the concatenation. We will clarify this in the revised version.
> -	The notations $\boldsymbol T$, $\boldsymbol U$ and $\boldsymbol V$ are specified in Section 1.1.
> -	The proximal operator for the Frobenius norm is indeed obtained by scaling the norm: $$\mathrm{prox}_{\gamma \lVert \cdot \rVert_F}( \boldsymbol Z ) = \frac{\boldsymbol Z}{\lVert \boldsymbol Z \rVert_F} \max ( \lVert \boldsymbol Z \rVert_F - \gamma, \ 0 ).$$ We will clearly explain this in the revised version.
>
> For the experiments,
> -	Figure 1(c) compares the RMS error between: root PCP with noise-INDEPENDENT parameters, and stable PCP with noise-DEPENDENT parameters. The fact that root PCP can achieve similar performance (RMS error) without knowing the noise standard deviation makes it a more practical approach than stable PCP.
> -	For numerical comparisons on real data, we added some preliminary results in Section C.1 of the supplement, which show comparable recovery performance between root PCP (with universal, noise independent parameters) and stable PCP (with noise dependent parameters) on noisy video data. We will include these results in the final version, and will also include similar comparisons across a larger set of videos.

---

### Official Review · Reviewer_PhKR · 2021-07-17

**Rating:** 7
**Confidence:** 4

**Summary:**

This work combines stable Principal Component Pursuit and Square Root Lasso and proposes Square Root Principal Component Pursuit for low-rank + sparse recovery problem. Compared to stable PCP, Square Root PCP has the advantage that the choice of the regularization parameter is noise-independent, so it is easier to be deployed in applications where the noise level is unknown. Theoretical analysis and experiments confirm the effectiveness of the proposed method.

**Limitations And Societal Impact:**

I don't see the limitations in this paper.

**Main Review:**

Overall, this is a well written paper. It contains theoretical analysis that with a single, universal choice of the regularization parameter, the low-rank matrix and the sparse matrix can be stably recovered. ADMM is applied to solve Square Root PCP with details such as initializations and stopping criteria. This paper also contains experiments in different settings and they confirm the effectiveness of the proposed method.

I am also satisfied with the improvements made for this resubmission. I have only one comment:

There are some existing works using ADMM to solve Square Root Lasso and it is worth citing them in this paper:
- The flare Package for High Dimensional Linear Regression and Precision Matrix Estimation in R by Xingguo Li et al.
- Square-Root lasso with nonconvex regularization: An ADMM approach by Xinyue Shen et al.

**Time Spent Reviewing:**

6 hours

---

> ### Author Response · Authors · 2021-08-10
> **Response to Reviewer PhKR**
>
> Thank you for the positive comments! We appreciate the suggested references regarding the application of ADMM to square root Lasso, and will cite and discuss these papers in the revised version.

---

### Author Response · Authors · 2021-08-11
**Additional Changes to the Paper**

We will also make the following additional changes to the paper:
-    Statement of Thm 2.1: remove the requirement that $\delta \ge \lVert \boldsymbol Z_0\rVert_F$, and change (2.1) to $\lVert \hat{\boldsymbol X} - \boldsymbol X_0 \rVert_F \le \sqrt{80n_1 n_2+1} \cdot \lVert \hat{\boldsymbol L}+\hat{\boldsymbol S}- \boldsymbol L_0-\boldsymbol S_0 \rVert_F \le \sqrt{80n_1 n_2+1} \cdot (\delta+\lVert \boldsymbol Z_0 \rVert_F)$.
-    Bullet point on line 165: Although the error bound for StablePCPc (Theorem 2.1) is not tight, it does show some structural properties of the solution, namely that it bounds the norm of $(\widehat{\boldsymbol{L}}, \widehat{\boldsymbol{S}}) - (\boldsymbol L_0, \boldsymbol S_0)$ by the norm of $\widehat{\boldsymbol{L}} + \widehat{\boldsymbol{S}} - \boldsymbol L_0 - \boldsymbol S_0$.
-    Statement of Thm 1.1: change the right-hand-side of (1.4) from $560 \sqrt{n_1 n_2} \lVert \boldsymbol Z_0 \rVert_F$ to $208 \sqrt{n_1 n_2} \lVert \boldsymbol Z_0 \rVert_F$.
-    Proof of Thm 1.1:
--	(A5): $ \lVert (\widehat{\boldsymbol{\delta}_L}, \widehat{\boldsymbol{\delta}_S}) \rVert_F \le 13 \sqrt{n_1n_2/2} \cdot \lVert \widehat{\boldsymbol{\delta}_L} + \widehat{\boldsymbol{\delta}_S} \rVert_F $.
--	Combining (A4) and (A5), we get $ \lVert \widehat{\boldsymbol{\delta}_L} \rVert_F + \lVert \widehat{\boldsymbol{\delta}_S} \rVert_F \le  13 \sqrt{n_1n_2} \cdot \lVert \widehat{\boldsymbol{\delta}_L} + \widehat{\boldsymbol{\delta}_S} \rVert_F \le 1/6 ( \lVert \widehat{\boldsymbol{\delta}_L} \rVert_F + \lVert \widehat{\boldsymbol{\delta}_S} \rVert_F ) + 520/3 \sqrt{n_1n_2} \lVert \boldsymbol Z_0 \rVert_F $.
--	The final bound becomes $208\sqrt{n_1n_2} \lVert \boldsymbol Z_0 \rVert_F$.

---

### Decision · Program_Chairs · 2021-09-27

**Decision:**

Accept (Poster)

**Comment:**

The paper studies the low-rank matrix recovery problem from observations corrupted by noise and sparse outliers. It proposes a framework called square root principal component pursuit, which is a combination of stable principal component pursuit and square root Lasso. The proposed technique has the advantage that the choice of the regularization parameter is noise-independent, so it is easier to be deployed in applications where the noise level is unknown. The main concern is about the estimation error bound obtained in the paper which is sub-optimal by a dimension-dependent factor. More explanation on why this sub-optimality occurs compared to prior works should be provided.